# Mortality for Lung Cancer among PVC Baggers Employed in the Vinyl Chloride Industry

**DOI:** 10.3390/ijerph19106246

**Published:** 2022-05-20

**Authors:** Paolo Girardi, Fabiano Barbiero, Michela Baccini, Pietro Comba, Roberta Pirastu, Giuseppe Mastrangelo, Maria Nicoletta Ballarin, Annibale Biggeri, Ugo Fedeli

**Affiliations:** 1Department of Environmental Sciences, Informatics and Statistics, Ca’ Foscari University of Venice, 30172 Venice, Italy; paolo.girardi@unive.it; 2Department of Medical Area (DAME), University of Udine, 33100 Udine, Italy; fabiano.barbiero@uniud.it; 3Department of Statistics, Computer Science, Applications, University of Florence, 50134 Florence, Italy; michela.baccini@unifi.it; 4Department of Environment and Health, Istituto Superiore di Sanità, 00161 Roma, Italy; pietro.comba@iss.it; 5Department of Biology and Biotechnology “Charles Darwin”, Sapienza Rome University, 00185 Roma, Italy; roberta.pirastu@gmail.com; 6Section of Occupational Medicine, Department of Cardiac, Thoracic, Vascular Sciences & Public Health, University of Padova, 35128 Padova, Italy; giuseppe.mastrangelo@unipd.it; 7Occupational Health Service, Local Health Authority 3, Veneto Region, 30174 Venice, Italy; nicoletta.ballarin@aulss3.veneto.it; 8Unit of Biostatistics, Epidemiology and Public Health, Department of Cardiac, Thoracic, Vascular Sciences and Public Health, University of Padova, 35128 Padova, Italy; annibale.biggeri@unipd.it; 9Epidemiological Department, Azienda Zero, Veneto Region, 35132 Padova, Italy

**Keywords:** lung cancer, PVC exposure, multiple imputation, mortality study

## Abstract

Vinyl-chloride monomer (VCM) is classified as a known carcinogen of the liver; for lung cancer, some results suggest a potential association with polyvinyl chloride (PVC) dust. We evaluated the relationship between lung cancer mortality and exposure as PVC baggers in a cohort of workers involved in VCM production and polymerization in Porto Marghera (Venice, Italy) considering both employment status and smoking habits. The workers were studied between 1973 and 2017. A subset of them (848 over 1658) was interviewed in the 2000s to collect information about smoking habits and alcohol consumption. Missing values were imputed by the Multivariate Imputation by Chained Equations (MICE) algorithm. We calculated standardized mortality ratios (SMR) and 95% confidence intervals (95% CIs) using regional reference rates by task (never, ever, and exclusively baggers) and by smoking habits. Mortality rate ratios (MRR), adjusted for age, calendar time, time since first exposure, and smoking habits, were obtained via Poisson regression using Rubin’s rule to combine results from imputed datasets calculating the fraction of information due to non-response. Lung cancer mortality was lower than the regional reference in the whole cohort (lung cancer SMR = 0.92; 95% CI 0.75–1.11). PVC baggers showed a 50% increase in lung cancer mortality compared to regional rates (SMR = 1.48; 95% CI 0.82–2.68). In the cohort analyses, a doubled risk of lung cancer mortality among PVC baggers was confirmed after adjustment for smoking and time-dependent covariates (MRR = 1.99, 95% CI 1.04–3.81). Exposure to PVC dust resulting from activity as bagger in a polymerization PVC plant was associated with an increase in lung cancer mortality risk after adjustment for smoking habits.

## 1. Introduction

Polyvinyl chloride (PVC, CAS No: 9002-86-2) is a synthetic resin made from the polymerization of vinyl chloride monomer (VCM, CAS No: 75-01-4). The occupational exposure to VCM peaked in the US and Europe in the 1950s but lasted until the 1970s [1]. VCM was classified by the International Agency for Research on Cancer (IARC) as a certain carcinogen, group 1: angiosarcoma of the liver and hepatocellular carcinoma are associated with exposure to VCM [1]. Additionally, suggestive evidence was found for connective and soft tissue malignant neoplasms, but weak evidence was found for the association with other cancers [1]. Concerning the increased risk of lung cancer, some controversial results lead to a lack of overall evidence for relations with vinyl chloride, despite some suggestions reporting, among PVC packers and baggers, an increased risk of lung cancer with cumulative exposure to VCM [1,2,3,4]. Polyvinyl chloride is classified as a possible carcinogen (class 3) by IARC. Inhaled polyvinyl chloride dust (in particular with an aerodynamic diameter of less than 5 mm) may remain in the pulmonary interstitium for years, gradually releasing residual VCM, which may account for the neoplastic transformation of an epithelial cell [5]. Due to the residual presence of VCM and other additives, the European classification, labeling and packaging (CLP) regulation reported as the PVC is one among the plastic polymers with the highest health hazard (highest hazard score 5) [6].

Some epidemiological data in PVC baggers suggest the role of PVC dust as a ‘promoting’ carcinogen operating in the last stages of lung carcinogenesis. [7,8]. On the contrary, in baggers from the English cohort of 5498 VCM workers [9], the relative risk of lung cancer was lower than unity. More recently, in the multicenter study, which included 12,700 workers from 19 VCM/PVC plants in four European countries, among individuals who only worked as PVC packers, the risk of lung cancer significantly increased with cumulative concomitant exposure to VCM [4]. In an Italian cohort of 1658 workers from a VCM/PVC plant in Porto Marghera (Venice, Veneto Region, Italy), the SMR for lung cancer was 1.73 (90% confidence interval 0.93–3.21) among ‘only baggers’; the ratio between SMR for ‘only baggers’ and that for ‘never baggers’ was 2.31 (90%CI: 1.15–4.61) [10]. In a re-analysis of mortality data from the same Italian plant, with respect to the reference group (technicians and clerks), the lung cancer rate ratio was 3.13 (95% CI 0.96–10.28) in PVC baggers [11]. Lastly, in a case–control study nested in the same Porto Marghera cohort, 38 patients with a histological diagnosis of lung cancer were compared with 224 controls without cancer. A logistic regression analysis showed an increase of 20% (odds ratio: 1.20; 95% CI: 1.07–1.35) in the risk of lung cancer for each additional year of work as a PVC packer, taking into account age and smoking [8]. The PVC dust concentration in this plant was an order of magnitude higher than that reported by Jones in the English cohort [9]. Mortality figures for the Italian cohort of Porto Marghera were updated until 2017 [12], but the effect of smoking habit was never considered as a confounding factor in the analyses. The objective of the present study is to assess the risk of lung cancer among the baggers in the Porto Marghera cohort, taking into account the information on smoking habits provided in an interview conducted on a subsample of workers, which was never considered in previous analyses.

## 2. Materials and Methods

This is a retrospective cohort study based on the mortality follow-up of 1658 workers in a VCM/PVC plant in Porto Marghera (Italy). Recruitment of the cohort and evaluation of causes of death were previously described [10,12,13]. VCM exposure by task and calendar period was estimated with a job-exposure matrix (JEM), which was built using estimates of exposure levels provided by the companies’ production process, as well as IARC information in the study of European workers exposed to VCM [4,14]; the JEM was then applied to individual work histories to compute the cumulative exposure in parts per million (ppm) years to VCM. However, individual cumulative exposure to PVC dust was unknown. Here, we considered the job as a proxy of individual PVC dust cumulative exposure. We classified it as follows: (i) never baggers (*n* = 1450); (ii) sometimes baggers: bagging together with other jobs (*n* = 120); (iii) exclusively baggers: subjects who worked exclusively as baggers (*n* = 88).

The start date of follow-up was 1 January 1973; the end was the date of death, end of residence in the Veneto region or 31 December 2017, whichever came first. Individual vital status was assessed through municipal registry offices, and causes of death through the National Institute of Statistics (ISTAT). Causes of death were coded according to the International Classification of Diseases 9th Edition (ICD-9).

Mortality figures for Porto Marghera cohort were updated until 2017 [12], but the confounding effect of smoking habits was never considered. Additional information on confounders such as smoking and alcohol consumption (alcohol was considered in the present analysis only in the missing data imputation phase) was retrieved from the interviews of 848 workers of the cohort (51.2%) conducted in 2004 by personnel from the Department of Statistics of the University of Florence using Computer-assisted telephone interviewing (CATI) (91.6%) and by health visits (8.4%). Through the CATI, it was possible to collect information on the workers not alive on 31 December 2003, by contacting their closest relatives.

To complete the missing values for the non-interview subjects’ records, we used the Multivariate Imputation by Chained Equations (MICE) algorithm [15], under a missing at random (MAR) assumption [16]. The variables of interest with missing values were related to past smoking habits and alcohol consumption. The missing data pattern was almost monotonous (out of the 832 subjects reporting missing information about smoking habits, 810 had both variables missing). We treated smoking habits in a block of two variables (smoking habits: no/former/current; the amount of cigarettes/day), while alcohol consumption was considered as continuous (gr alcohol/day). Before starting the imputation process, a coherence control was performed between smoking habit and the amount of cigarettes/day smoked: 7 smokers who reported a consumption of 0 cigarettes/day were recoded as “no smokers”. In order to consider non linearity and interactions, we performed the imputations within the MICE algorithm by classification and regression trees (CART). For the variables to be imputed, we considered the following as explanatory variables in the CART (Appendix A): (a) age of first exposure; (b) year of first exposure; (c) duration of the working activity in the plant; (d) time since leaving of work to the previous follow-up date (31 July 1999) [10] or death (if it happened before); (e) life status at the previous follow-up date (at 31 July 1999) (0 = alive, 1 = died for lung cancer, 2 = died of cardiovascular diseases, 3 = died of liver cancer, 4 = other causes); (f) time from July 31, 1999 to the last follow-up date (date of the last follow-up or death, if a new death occurred) or 0 if the subject was dead before; (g) life status at the last follow-up (0 = alive or dead before the interview, 1 = died for lung cancer, 2 = died for cardiovascular diseases, 3 = died for liver cancer, 4 = died for other causes); and (h) task as a bagger (never, ever, exclusive). The temporal variables were not categorized, since CART model operates optimal splits on the explanatory variables. It should be pointed out that, using CARTs, imputations outside the range of the observed values are avoided, being the imputed values drawn from the observed values. Thirty iterations of the algorithm were deemed sufficient to achieve convergence, since the missing data pattern was approximately monotonous. We produced 30 multiple imputed data sets and performed on each of them all of the planned analyses.

To make external comparisons, we calculated standardized mortality ratios (SMR) along with their 95% confidence intervals (95% CIs) [17] using as references the mortality rates by sex, age, and calendar period of the Veneto region for the years 1970–2015, provided by ISTAT. The rates of last available reference period were extended up to 2017. The SMRs were calculated as the ratio of the observed number of deaths (O) in the cohort to the expected one (E) on the basis of the regional reference. The SMRs for lung cancer were estimated for the entire cohort by job as a bagger (never, ever, exclusively); by smoking habit, recorded in three categories (no smoker, <15 cigarettes; >15 cigarettes); and by latency (≤10, 11–20, 21–30, >30 years). With the aim of conducting internal comparisons, mortality rate ratios (MRR) were estimated by specifying Poisson regression models on the lung cancer deaths, setting the person years as offset variable. The Poisson models included as explanatory variables: job (never, ever, exclusively bagger), smoking habit (no smoker, <15 cigarettes; ≥15 cigarettes), calendar period (≤1980, 1981–1990, 1991–2000, 2001–2010, >2010), age (≤50, 51–60, 61–70, 71–80, >80) and latency (≤10, 11–20, 21–30, >30). An additional analysis for work duration as a bagger (no work, <3 years; ≥3 years) was conducted.

The point estimates and the standard errors of the SMRs by smoking habit and of the MRRs obtained on the 30 imputed data set were combined by means of the Rubin’s rule [18]. In the Poisson regression model, in order to determine the impact of missing data from smoking on the efficiency of risk estimates, we calculated the fraction of missing information (γ) [19]. All analyses were carried out using R software 4.2. The *mice* package was used for the multiple imputation procedure [15].

## 3. Results

As reported in Table 1, the cohort was predominantly formed by workers employed as polymer and compounds operators (40% and 25%), hired mainly in the period 1961–1972, with a low fraction of employees working as baggers (7.2% and 5.3% ever and exclusively, respectively). The life status at the last follow-up reported a total of 731 deceased: lung cancer was the second cause of death (*n* = 100) after cardiovascular diseases (*n* = 172).

The distribution of the study variables by available/imputed data on smoking habit status is shown in Appendix A. Overall, among the 826 workers for whom the smoking habit information was available, 36.1% were no smokers, 46.6% were current smokers, 17.3% were former smokers (Figure 1). The average number of cigarettes/day was equal to 10.2. Among the deceased who had information on smoking habits (*n* = 303), all lung cancer deaths except one were current (31.7%; 20/63) or former smokers (66.7%; 42/63), while other deaths reported a lower percentage of former smokers (17.5%; 42/240), but higher percentage of current smokers (54.6%; 131/240). Among all lung cancers (*n* = 100), 11% and 8% were ever and exclusively employed as baggers, percentages similar to those reported by non-lung cancer deaths (9.7% and 7.0% for ever and exclusively bagger, respectively). Among baggers, the work duration in this task was an average of 4.97 years, higher among those employed exclusively as baggers (average duration: ever 3.26 years vs. exclusively 7.31 years; *t*-test: *p* < 0.001).

In the imputation results, a slight difference in the smoking habit distribution was found between the complete cases and the imputed data sets, where the percentage of active smokers was higher, as well as the average number of cigarettes smoked per day (Figure 1). The cohort accrued a total of 61,736 person years at risk (37.2 years the mean period of follow-up for each worker). A moderate increase in lung cancer mortality among exclusive bagger workers was found compared to the regional reference (SMR: 1.48, 95% CI: 0.82–2.68). The estimated SMRs were 0.36 (95% CI: 0.18–0.71) among no smokers, 0.60 (95% CI: 0.38–0.95) for workers in the category < 15 cigarettes/day, and 1.99 (95% CI: 1.53–2.58) among workers in the category ≥ 15 cigarettes/day (Table 2). Internal MRR analysis, adjusted for smoking habit and time-dependent variables (age, period, and latency), confirmed increased mortality from lung cancer among workers employed exclusively as baggers compared to those never employed in that task. The estimated MRR on the subset of complete cases was equal to 2.50 (95% CI: 1.09–5.03). This difference remained evident also considering the entire cohort of workers (MRR: 1.99, 95% CI: 1.04–3.81) (Table 3). The analysis for work duration as a bagger reported a tendency towards an increased risk among those at work more than 3 years (MRR: 1.72; 95% CI: 0.88–3.37), compared to those who never worked as a bagger (Appendix A). The effect of the fraction of missing information on smoking habit risk estimates due to missingness was moderate (41.9% and 37.9% for 1–15 cigarettes, and ≥15 cigarettes), while those related to work as a bagger were below 5%.

## 4. Discussion

The study confirmed some evidence provided by previous studies that showed an increase in the risk for lung cancer among PVC baggers [8,10]. Such increased risk of lung cancer was found more frequently among heavily exposed cohorts and workers employed in older plants [2,20,21] compared to more recent cohorts, as a result of the registered decrease in exposure levels in the 1980s [1]. However, most of the studies did not report information about smoking habits, which remained the most significant cause of lung cancer among the general population. The present study included information on smoking in the analysis of a group of workers. Considering a multivariate imputation process, we obtained information on smoking habits for the entire cohort. Eventually, despite the scarce number of baggers in this occupational cohort, we found a higher risk of lung cancer among only baggers, after controlling for smoking habits in the Poisson regression model. The adjustment for smoking habits strengthens the results with respect to the previous study [10], excluding a potential important source of bias. A tendency towards an increased lung cancer risk with increasing duration of work as a bagger (including both ever and exclusive baggers) could be observed. It must be remarked that this is the first cohort study in PVC workers with lung cancer risk estimates adjusted for smoking habits. The estimated fraction of missing information on smoking habit risk was acceptable, and the achieved efficiency of the related risk estimates remained high, while the influence on the risk related to the task of being a bagger was limited [18].

The pathogenic role of PVC was shown by some laboratory studies to reveal the link between exposure to PVC dust and both non-malignant and malignant lung disease. Despite the low reactivity, the number of surface area atoms per unit mass was high for PVC dust, greatly enhancing surface area for chemical reactions with bodily fluids and tissue in direct contact, resulting in persistent inflammation that led to pulmonary fibrosis or even carcinogenesis [22]. The idea of PVC dust being a toxic agent is supported by multiple lines of evidence, such as the presence of PVC particles inside macrophages from human patients [23] and the onset of pathomorphological alterations (pulmonary inflammation and damage similar to silica-induced inflammation) in rat lungs 2 days after instillation with PVC particles. [24]. In addition, enzymatic and morphologic alterations in rat lungs were studied at different time intervals up to 180 days after a single intratracheal administration of 25 mg of polyvinyl chloride dust. Pathologically, the pulmonary response was in the form of acute inflammatory changes during the early stages of dust burden, followed by the development of granulomatous lesions containing small amounts of stromal elements [25].

Along with several additives (initiators, emulsifiers, protective colloids, etc.), PVC dust might contain residual VCM. However, as reported for liver diseases, P450-2E1 protein is pivotal for activating an indirect carcinogen route for VCM; the observed increased risk may be explained by its hypomethylation in lung tissues confirmed by a small study on a minority of the individuals of the considered cohort [26]. In addition, the lung can be the target of the effects of PVC dust. Considering animal models, no excess cancer was found in long-term inhalation studies. However, a certain grade of toxicity was reported by several studies. In rats exposed to PVC, titanium dioxide, and powdered iron, using a directed-flow nose-only mode of exposure for 6 h/d and 5 days/w for 7 months, histological examinations of the lungs revealed the accumulation of particle-laden macrophages, interstitial inflammation, proliferation of connective tissue, and lymphoid hyperplasia. These effects were more marked in rats exposed to PVC [27]. In another study, rats were subjected to intratracheal doses of PVC particles for three weeks. In all treated animals, the analysis of the bronchoalveolar lavage fluid revealed an increase in neutrophils and macrophages and an increased CD4/CD8 ratio in lymphocytes, while histopathological examinations revealed a mild inflammation of the lungs. The latter findings, in combination with the increased CD4/CD8 ratio, indicate initial alveolitis [28].

In humans, 20 cases of PVC-induced pneumoconiosis were found in the PVC production plant of the present study: all workers were exposed to high PVC dust levels for at least five years, and among them, a small percentage of cases reported slightly restrictive respiratory function impairments [29]. It must be remarked that in the drying, sacking and blending departments of the plant, PVC dust concentrations were over 10 mg/m^3^ of the total dust in about 60% of the samples, and in the samples taken, particles reaching the alveolar surface with diameters of 1 μm to 6 μm constituted 4.5 to 30.9% of total dust weight [29]. Likewise, among 171 Chinese and Malay PVC compounding workers compared with a reference group, workers with high cumulative PVC dust exposure had a higher prevalence of radiological profusion of small opacities and a small degree of lung function impairment [30].

Since events that trigger carcinogenic transformation (persistent alveolar inflammation, alveolar macrophage activation, release of cytokines, chemokines, and growth factors) were reported for PVC dust, a carcinogenic pulmonary effect can be expected to take place after long-term exposure to high levels of this dust. The German Commission for the Investigation of Health Hazards of Chemicals in the Work Area has recently reevaluated polyvinylchloride [31]. The German Committee (‘MAK-Commission’) considers the mode of action of tumor development in its classification scheme. In comparison to other bio-persistent granular dust, PVC dust was classified as carcinogen Category 4, and a MAK value of 0.3 mg/m^3^ × material density was established for respirable dust. This value is valid for PVC containing no additives with a monomer content of <1% [31]. The Group 4 category includes carcinogens with a non-genotoxic mode of action, for which no significant excess of cancer risk is expected at exposures up to MAK values [32].

The main limitation of the present study was the limited sample size, which affected the power of risk estimates for duration of exposure as well as other combinations with the latency or work period. Information about previous workplaces exposing to PVC/VCM was not available, but since the young age at the hiring of the cohort, the potential exposure underestimation or misclassification was limited. In addition, a potential exposure to asbestos in the petrochemical plant cannot be excluded, but this latter exposure is unlikely to be differential between baggers and non-baggers.

## 5. Conclusions

Excluding a potentially important source of bias, the adjustment for smoking strengthened the results of previous studies showing an increased risk for lung cancer among PVC baggers. Long-term exposure to high levels of PVC dust might promote pulmonary carcinogenesis through persistent alveolar inflammation, alveolar macrophage activation, and release of growth factors. Future collaborative research collecting information from different cohorts of workers employed as baggers or in other tasks entailing high levels of PVC dust exposure is warranted.

## Figures and Tables

**Figure 1 ijerph-19-06246-f001:**
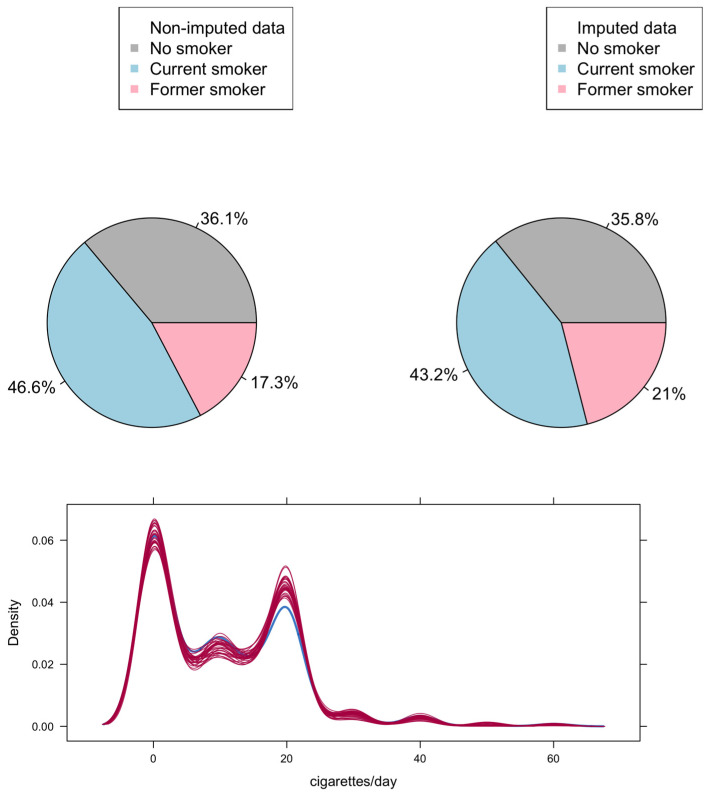
Complete cases data set and the imputed data sets: distribution of smoking habit (average of the imputed) and of number of cigarettes/day (blue line complete, red lines imputed).

**Table 1 ijerph-19-06246-t001:** Main characteristics of the cohort as a whole and from interviews.

	Overall, *n* = 1658 ^1^	Interviewed, *n* = 848 ^1^	Non Interviewed, *n* = 810 ^1^
**Age at start of follow-up (years)**	36 (28, 42)	36 (31, 41)	34 (26, 42)
**Year of hire**	1968 (1961, 1972)	1967 (1961, 1972)	1970 (1961, 1973)
**Task**			
White collar	23 (1.4%)	12 (1.4%)	11 (1.4%)
Monomer operator	262 (16%)	125 (15%)	137 (17%)
Polymer operator	663 (40%)	338 (40%)	325 (40%)
Compounds operator	416 (25%)	219 (26%)	197 (24%)
Technician	294 (18%)	154 (18%)	140 (17%)
**Task as a bagger**			
Never	1450 (87%)	745 (88%)	705 (87%)
Ever	120 (7.2%)	62 (7.3%)	58 (7.2%)
Exclusive	88 (5.3%)	41 (4.8%)	47 (5.8%)
**Work duration (years)**	12 (6, 19)	14 (9, 21)	11 (4, 17)
**Cumulative VCM exposure**			
0–733 ppm years	1214 (73%)	605 (71%)	609 (75%)
734–2388 ppm years	218 (13%)	108 (13%)	110 (14%)
2379–5187 ppm years	148 (8.9%)	89 (10%)	59 (7.3%)
≥5188 ppm years	78 (4.7%)	46 (5.4%)	32 (4.0%)
**Status at last follow-up (2017)**			
Alive	927 (56%)	523 (62%)	404 (50%)
Deceased from lung cancer	100 (6.0%)	66 (7.8%)	34 (4.2%)
Deceased from cardiovasc. dis.	172 (10%)	76 (9.0%)	96 (12%)
Deceased from liver cancer	57 (3.4%)	32 (3.8%)	25 (3.1%)
Deceased from other causes	402 (24%)	151 (18%)	251 (31%)
**Type of interview** (*n* = 848)			
CATI	-	777 (92%)	-
Clinical interview	-	71 (8.4%)	-
**Smoking habit** (*n* = 826)			
No smoker	-	291 (36%)	-
Current smoker	-	392 (47%)	-
Former smoker	-	143 (17%)	-
**Smoking consumption** (cigarettes/day) (*n* = 826)	-	10 (0, 20)	-
Alcohol habit (*n* = 824)			
No	-	147 (18%)	-
Yes	-	677 (82%)	-
**Alcohol consumption** (gr alcohol/day) (*n* = 824)	-	25 (12, 38)	-

^1^ Median (IQR); *n* (%)

**Table 2 ijerph-19-06246-t002:** Observed deaths (O), person years (PYs), expected deaths (E) and standardized mortality ratios (SMR) for lung cancer with 95% confidence intervals (95% CI) for the whole cohort, by latency, by task as a bagger and by smoking habit.

	O	PYs	E *	SMR	95% CI
**Overall**	100	61,736	109.2	0.92	0.75–1.11
**Latency (years)**					
≤10	8	16,372	8.08	0.99	0.49–1.98
(10–20]	19	15,668	21.1	0.90	0.58–1.41
(20–30]	33	14,203	31.3	1.05	0.75–1.48
>30	40	15,472	48.8	0.82	0.60–1.12
**Task as a bagger**					
Never	81	54,648	92.9	0.87	0.70–1.08
Ever	8	4197	8.95	0.89	0.45–1.79
Exclusive	11	2891	7.41	1.48	0.82–2.68
**Smoking habit ****					
Never	14.6	22,704	39.7	0.36	0.18–0.71
<15 cig/day	22.9	21,516	38.2	0.60	0.38–0.95
≥15 cig/day	62.5	17,517	31.4	1.99	1.53–2.58

* Expected deaths calculated using regional rates by sex, age and calendar period. ** Person years, and observed and expected cases are reported as the mean of cases across the 30 imputed data sets.

**Table 3 ijerph-19-06246-t003:** Adjusted * multivariate mortality rate ratios (MRR) for lung cancer estimated by Poisson regression and 95% confidence intervals (95% CI) for the complete cases and for the imputed datasets. The value of γ represents the fraction of information missing due to nonresponse.

**Complete Cases (*n* = 848)**		**MRR**	**95% CI**	
Task as bagger	Never (reference)	1.00		
	Ever	1.31	(0.50–2.84)	
	Exclusive	2.50	(1.09–5.03)	
Smoking habit	No smoker (reference)	1.00		
	<15 cigarettes	4.53	(1.46–19.8)	
	≥15 cigarettes	23.2	(8.53–95.6)	
**Entire Dataset (*n* = 1658)**		** MRR **	**95% CI**	** γ **
Task as Bagger	Never (reference)	1.00		
	Ever	1.05	(0.50–2.20)	1.85%
	Exclusive	1.99	(1.04–3.81)	3.34%
Smoking habit	No smoker (reference)	1.00		
	<15 cigarettes	1.67	(0.69–4.04)	41.9%
	≥15 cigarettes	5.69	(2.70–12.0)	37.9%

* Adjusted for 10-year age categories (≤50, 51–60, 61–70, 71–80, >80), calendar period (≤1980, 1981–1990, 1991–2000, 2001–2010, >2010), and latency (≤10, 11–20, 21–30, >30).

## Data Availability

The data presented in this study are available on request from the corresponding author. The data are not publicly available.

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
