# Peer review of "Mortality for Lung Cancer among PVC Baggers Employed in the Vinyl Chloride Industry"

_ijerph, 2022, doi:10.3390/ijerph19106246_

Round 1

Reviewer 1 Report

The article entitled „Mortality for lung cancer among PVC baggers employed in the vinyl chloride industry” is well conducted which focused under lung cancer issue among PVC industries after adjustment for smoking habits. The article was based on the retrospective cohort study, on the mortality follow-up of 1658 workers in a VCM/PVC plant in Italy (Porto Marghera).

Special comments:

  1. In the Introduction section I would add information concerning CLP classification of VCM and PVC.
  2. In the Results section, I was wondering about the previous wokplace of workers in exposure to PVC/VCM? Why is it not included in the table/text?
  3. Table 1. Deceased for other causes. Have the Authors noticed bladder cancer?

Author Response

Please find notes in the attached file

Reviewer 2 Report

The manuscript entitled “Mortality for lung cancer among PVC baggers employed in the vinyl chloride industry” aims to assess the risk of lung cancer among the baggers in the Porto Marghera cohort, taking into account the information on smoking habits provided in an interview conducted on a subsample of workers. The novelty of this topic is not high, but the importance is. The manuscript is well prepared. The introduction is informative, the results are comprehensible, and the results support the conclusions. Still, I think that discussion can be improved. There are many articles on this topic with newer date, so the author should discuss regarding recent literature. Also, the papers already cited are a bit out of date. The conclusion is rather short and should be given in more detail.

Author Response

please find notes in the attached file

Reviewer 3 Report

The research article by Girardi and colleagues evaluated the relationship between lung cancer mortality and exposure as PVC bagger in a cohort of workers involved in VCM production and polymerization in a VCM/PVC plant located in Italy combining both the employment status and the smoking habit. Considering that the literature on the subject is quite heterogeneous and presents some methodological limitations, it cannot be concluded whether there is an association between occupational PVC bagger or packer exposure to inhaled dust or their residual VCM and the mortality or incidence for lung cancer among the workers employed in the vinyl chloride industry. In their previous studies of Italy large petrochemical plant, the lung cancer OR increases by 20% for each extra year of work, when the influence of age and smoking habits is controlled (Mastrangelo et al. 2003). The higher than two-fold increases were found for liver cirrhosis (RR = 2.8) among autoclave workers, for lung cancer (RR = 3.13) among PVC baggers and for liver cancer (RR = 2.46) among PVC compound workers (Gennaro et al. 2008). In this study, the author given the certain results for PVC baggers showed a 50% increase in lung cancer mortality compared to regional rates (SMR = 1.48). In within cohort analyses, a doubled risk (MRR = 1.99) of lung cancer mortality among PVC baggers was confirmed after adjustment for smoking and time-dependent covariates. Most of my concerns are related to the smoking habit, PVC-contacted duration, individual cumulative exposure to PVC dust, and the interpretation of their finding. Previous studies reported the risk of lung cancer after the potential confounders (including smoking habits). The results and discussion on their findings seems not to carry full conviction for the novelty of this study from the previous studies. The detail information of statistic method and employed time to cancer incidence estimation need to provide. Nevertheless, there are some problems existing in present study, the authors should make some explanations and modifications.

Major Comments

  1. It is not clear to realize the novelty of this study from the previous studies on the risk for lung cancer among PVC workers. Most studies put smoking habits as the confounders and controlled it in their statistical models. The objective of the study is to assess the risk of lung cancer among the baggers in the Porto Marghera cohort, taking into account the information on smoking habits. The author took the smoking habits as an explanatory variables and evaluated separately the risks for the workers who employed for PVC task or smoking habits in their models. The risk for the workers with both variables of specific PVC task and smoking habit need to clarify in this study.
  2. The characteristics of the groups (lung cancer and non-lung cancer) with the variables of specific PVC task and smoking habit need to provide.
  3. The detail information of statistic method and employed time to cancer incidence estimation need to provide. Although the author reported the lack information of individual cumulative exposure to PVC dust, the PVC-contacted duration of the worker’s time activity could provide the important information to realize the occupational PVC exposure on the risk of lung cancer.
  4. The author should consider to discuss their findings on the risk of lung cancer via occupational exposure to PVC dust according to their results in Tables. The discussion suggest to re-write, following the significant results and non-significant results related to their objective.
  5. Table 1: “Cumulative CVM exposure” should be Cumulative VCM exposure. How did the estimation of “Cumulative VCM exposure”?
  6. The essential weakness of the research was not the small sample size (n= 1,658, follow-up), but no potential confounders included in the models was the problem. The authors should discuss this limitation.
  7. There are many mechanisms involved in these connections on the risk of lung cancer but not assayed in the present study. The author should edit carefully in their conclusion.

Author Response

please find notes in the attached file

Round 2

Reviewer 3 Report

The author has clarified my comments.